# Electro-Magnetic Geophysical Dynamics under Conservation and Conventional Farming

**Alberto Carrera** [1], **Matteo Longo** [1,*], **Ilaria Piccoli** [1], **Benjamin Mary** [2], **Giorgio Cassiani** [2] **and Francesco Morari** [1]

1  DAFNAE Department, University of Padova, 35020 Legnaro, Italy
2  Geoscience Department, University of Padova, 35100 Padova, Italy
*  Correspondence: matteo.longo.2@unipd.it; Tel.: +39-049-8272837

**Abstract:** In the context of global warming, agriculture faces severe challenges such as water scarcity and soil erosion. Key to achieving soil sustainability is the choice of farming practices, the consequences of which are generally site-specific. In this study, the ability of Electrical Resistivity Tomography (ERT) and Electro Magnetic Induction (EMI) methods were assessed for monitoring the effects of conventional (CONV) and conservation (CONS) agricultural practices. The aim is to highlight differences in soil water distribution caused by both short- and long-term effects of the two different practices. Results demonstrated that both ERT and EMI provided sufficient information to distinguish between the effects of CONV and CONS, while traditional direct measurements, being punctual techniques, lacked sufficient spatial resolution. The ERT transects showed that the soil was much more homogeneous as a result of CONS practices, resulting in a higher sensitivity to changes in the water content. Conversely, due to the heterogeneous soil structure under CONV, water distribution was more irregular and difficult to predict. Similar patterns were also observed with the EMI surveys, with a strong link to spatial variability. Finally, we conclude that for CONV soil, the accessible water for the plant is clearly controlled by the soil heterogeneities rather than by the forcing atmospheric conditions. This study is a first step towards paving the way for more refined hydrology models to identify which soil parameters are key to controlling spatial and temporal changes in soil water content.

**Keywords:** agrogeophysics; ERT; EMI; proximal sensing; conservation agriculture

## 1. Introduction

The importance of soil structure for sustainable production is increasingly recognized because of its central role in plant growth, soil ecological functioning, and impacts on water and energy fluxes to/from the atmosphere [1,2]. However, as a product of the fragile interaction of soil biological and anthropic activity [3,4], soil structure and soil inner processes are difficult to measure, both in time and space, because of our limited capability of monitoring the soil profile at the necessary spatial scale and time-frequency. Indeed, measurements of relevant metrics often rely on time-consuming, invasive physical methods that can be only episodically performed.

Geophysical methods are non-invasive sensing techniques that can measure the physical (often electrical and mechanical) properties of the investigated media. They can be used either qualitatively by identifying anomalies and contrasts and thus spatial heterogeneities or quantitatively by relating them to properties and states of interest through physical relationships [5,6]. In agriculture, geophysical methods, such as electrical resistivity tomography (ERT) and electromagnetic surveys, are increasingly playing an important role in the characterization of soil spatial variability and hydrological processes, of primary interest for effective management and precision farming [7–9]. ERT has been used to monitor the spatial variability of soil physico-chemical properties [10–13] and the effects of different

agricultural practices (cover crops, compaction, irrigation, tillage, and fertilization) on soil water dynamics and crop yields [14]. ERT has become a standard tool in agricultural investigation due to its robust nature, proving its suitability for a range of tasks, such as delineating soil horizons, estimation of water content, and monitoring purposes [15–18]. Surveys are performed with multi-electrodes devices in order to obtain the electrical resistivity distribution of the subsoil in a 2D or 3D model. An array of dozens of electrodes is coupled with soil (either at the surface or buried in depth) to ensure galvanic contact with the ground. The apparent resistivities $\rho_a$ ($\Omega$m) of the subsoil are retrieved by injecting the current I (A) in two of the electrodes (current electrodes, called A-B) and by recording the potential difference $\Delta$V (V) that arises between two other electrodes (the potential ones, called M-N). The measurements are performed along the entire electrodes array, using a number of A-B, M-N configurations, generating a pseudo-section of apparent resistivities (i.e., the resistivity of an equivalent homogeneous subsurface producing the same $\Delta$V/I ratio) [19]. The spatial resolution of the survey is inversely related to the distance between electrodes: the smaller the spacing, the higher the resolution. The current penetration depends mainly on the electrical properties of the investigated media, the spacing and the configuration of electrode quadrupoles [20], but also on the acquisition sequence. The inversion process of the collected dataset finally produces an estimate of the real distribution of the electrical resistivity in the subsoil. This can be performed by using codes which, starting from a discretization grid, iteratively find the best subsoil resistivity model that minimize the misfit between the measured and the computed dataset, often also satisfying a-priori constraints such as smoothness [20,21].

Electromagnetic induction (EMI) methods are a proximal sensing approach that has increasingly been used for soil mapping and related precision farming purposes [22–26]. Note that ERT and EMI measure, albeit in different manners, the same physical property, i.e., electrical resistivity (or its reciprocal electrical conductivity). EMI has been widely applied in environmental applications [27] to rapidly map soil electrical conductivity (hereinafter EC) of the subsoil related to variations of, e.g., salinity [28,29], water content [30,31], soil texture [25,32], soil organic matter [33,34]. In agricultural soils, the resistivity range is commonly low ($10^{-1}$–$10^2$ $\Omega$m), making the response adequate to the resolution limits of EMI instruments (~0.1–1000 mS/m, www.gfinstruments.cz, accessed on 9 November 2022, 1 S/m = 1/$\Omega$m) [28,35,36]. This fact, added to EMI's ease of use and fast data collection over large areas, is the main reason why this technique is particularly popular in precision farming. Recently, the implementation of easy-to-use EMI inversion codes has led to a major step forward, allowing the user to build quasi-3D electrical conductivity models of the near-surface [37–39]. While EMI inversion cannot match the detailed results of ERT, EMI's capability to collect data over very large areas in a very short time largely overwhelms EMI limitations. ERT and EMI together make a perfect couple of techniques that can cover large areas with the necessary local detailed calibration.

Electrical and electromagnetic techniques have been widely and successfully used in the characterization of soil properties (bulk density and clay content) as well as state variables (soil salinity, water content, and water saturation), as they are sensitive, through the effect on bulk electrical conductivity, to porosity, pore water conductivity, saturation, grain mineralogy, and bulk density [7,40–42]. Some of these properties, such as texture, are stable in the long term. Others, such as soil moisture, depend on the forcing conditions and thus vary over time, demanding a time-lapse survey approach, where repeated measurements are conducted using the same spatial configuration. However, although recent developments have facilitated ERT and EMI acquisition by making it easier to perform, an inherent uncertainty exists in the identification of the dominant factors that influence EC variability at a specific site (such as soil moisture, temperature, and salinity).

Coupling proximal geophysical sensing with below-ground direct measurements can increase our understanding of the soil response [43,44]. Furthermore, coupling geophysically-based proximal measurements with meteorological data allows for a more complete understanding of the whole soil–plant–atmosphere continuum (SPAC) system. In this context,

a geophysical time-lapse approach can highlight with adequate space-time resolution the soil moisture changes induced by seasonal variations, precipitation, and root water uptake [41,45–47].

Consequently, geophysical surveying is particularly promising in the study of agronomic issues and practices [9]. Traditional farming approaches rely on soil tillage before seeding, including plowing and secondary tillage operations. Heavy machinery is required, on which are mounted operating tools that increase their weight even more. Plowing modifies the natural soil structure and biological activity. Support for conservation agriculture (CONS) is part of a broader view towards sustainable development where economic and environmental objectives can be met in order to "produce more with less". CONS is a system of agronomic practices that minimizes mechanical soil disturbance, maintains permanent soil cover (i.e., crop residues, cover crops), and prescribes crop rotation [48–50]. In recent years, CONS has received increasing attention as a solution to minimize soil threats caused by intensive agricultural systems (e.g., organic content depletion, microorganism habitat loss, compaction) [51,52]. However, awareness is required as CONS has shown both negative and positive effects on soil structure properties (e.g., bulk density, soil strength) depending on the local context [53–55]. Even though this practice has been applied worldwide for a few decades, CONS effects on soil water dynamics are still a subject of heated controversial debates.

The aim of this study was to investigate the potential of ERT and EMI to highlight soil water dynamics with adequate space and time resolution, with the specific goal of understanding the effects of CONV and CONS management practices. Both EMI and ERT surveys can highlight differences between conservation and traditional soil management regarding electro-magnetic properties as a function of soil structure and moisture content. To this end, these methods have been applied at different scales: (i) at a detailed-spatial resolution scale, where ERT measurements have been collected in time-lapse along the same profiles, (ii) at a lower-spatial resolution scale, but larger aerial coverage, where EMI surveys were collected to highlight spatial heterogeneities.

## 2. Materials and Methods

### 2.1. Site Description

Experimental treatments were established in 2010 in the Veneto Region (North Eastern Italy) at an experimental farm located in the low-lying Venetian plain (45° 2.9080N 11° 52.8720E, 2 m a.s.l.) (Figure 1). The monitoring activity was conducted in 2017–2018 season On soil classified as Hypocalcic Calcisol (WRB, 2006) with a silt-loam average texture (Table 1). The water table level ranged from about −250 cm in summer to −70 cm in winter from the land surface. The climate is sub-humid, with an annual mean rainfall of 673 mm, uniformly distributed throughout the year. Temperatures are lower in January (−0.2 °C minimum average) and higher in July (30.6 °C maximum average).

The experiment compared conventional (CONV) versus conservation (CONS) management systems. Prior to the CONS transition (before 2010), the conditions of both fields were similar in terms of BD [11] and SOC [56,57]. The CONS protocol followed a set of practices outlined in Measure 214, Submeasure 1, "Eco-compatible management of agricultural lands" of the Rural Development Programme (RDP) 2007–2013 supported by Veneto Region. Summarizing, it required no-tillage, cover crop usage, and crop residue retention on soil surface. Contrariwise, the CONV management system used traditional tillage practices: moldboard plowing (35 cm depth), crop residue incorporation, and disk-harrowing to a depth of 10 cm. The experimental fields were approximately 500 m long by 30 m wide. Until 2014, the 4-yr crop rotation consisted of wheat (*Triticum aestivum L.*), oilseed rape (*Brassica napus L.*), maize (*Zea mays L.*), and soybean (*Glycine max (L.) Merr.*). From 2014 onwards, a simplified, 3-yr crop rotation (wheat–maize–soybean) was applied. In CONS, cover crops were grown between the main crops. Until 2014, sorghum (*Sorghum vulgare Pers. var. sudanense*) was grown during spring–summer, and a mix of vetch (*Vicia sativa L.*) and barley (*Hordeum vulgare L.*) was grown during autumn–winter. In the following years,

only barley or winter wheat was grown in autumn–winter. In CONV, the soil remained bare in the time intervals between the main crops. In CONV, the base dressing fertilizer was applied 1–2 weeks before sowing; subsurface band fertilization was applied at sowing in CONS. In both systems, mineral fertilization was integrated by side-dressing in maize (one treatment) and wheat (two treatments). Cover crops received no additional fertilization. Pesticides were applied based on crop need and were the same for both treatments. The winter cover crop was suppressed with N-(phosphonomethyl) glycine; sorghum was suppressed by mechanical shredding. Throughout our experiment, CONV field remained bare while CONS was covered by cover crops.

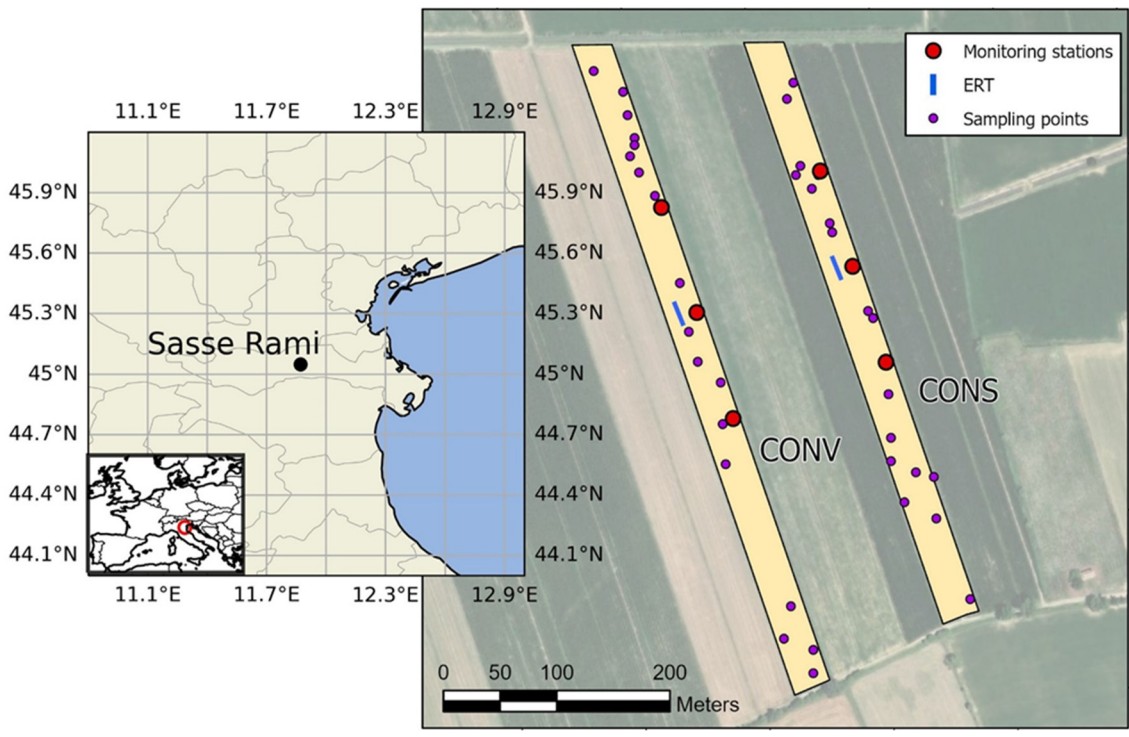

**Figure 1.** The Sasse-Rami experimental farm location and zoom to the CONV and CONS fields.

**Table 1.** Main soil physical and chemical characteristics (top 50 cm) of the experimental farm.

| Characteristic | Unit | Value |
|---|---|---|
| Sand | g 100 g$^{-1}$ | 18.4 |
| Silt | g 100 g$^{-1}$ | 57.8 |
| Clay | g 100 g$^{-1}$ | 23.8 |
| pH | - | 8.6 |
| Carbonate | g 100 g$^{-1}$ | 13.0 |
| Active carbonate | g 100 g$^{-1}$ | 3.0 |
| Organic carbon | g 100 g$^{-1}$ | 0.8 |
| Available P | mg kg$^{-1}$ | 6.0 |
| Exchangeable Ca$^{2+}$ | meq 100 g$^{-1}$ | 15.5 |
| Exchangeable Mg$^{2+}$ | meq 100 g$^{-1}$ | 1.4 |
| Exchangeable K$^{+}$ | meq 100 g$^{-1}$ | 0.2 |

## 2.2. Data Collection

Experiments occurred between December 2017 and April 2018, coupling soil characteristics measurements and geophysical surveys. During the course of the experiment, soil was bare in CONV while winter wheat cover crop was grown between tillering to boot stage in CONS. For each field, three monitoring stations were equipped with multi-sensor

probes (HD3510.2, Delta OHM, GHM GROUP, Selvazzano Dentro, Italy) which continuously monitored soil temperature (T, °C) and volumetric water content (VWC, %) at three depths (10, 30, and 55 cm). Prior to field installation, the soil moisture sensors, operating with a frequency domain reflectometry technique, were calibrated in the laboratory to an accuracy of ±3%. Data were recorded every 30 min and regularly monitored by a radio frequency wireless remote control system using ISM (Industrial, Scientific, and Medical) radio bands. The system connected the monitoring probes to a weather station (Delta OHM, GHM GROUP, Selvazzano Dentro, Italy) via GSM technology. The weather station was equipped with a thermometer, hygrometer, anemometer, pyranometer, and rain gauge. Five time-lapse geophysical surveys—i.e., ERT and EMI—were conducted on 15/12/2017, 05/02/2018, 14/03/2018, 26/03/2018, and 26/04/2018. On the same date as the last survey, soil cores were collected with a hydraulic sampler (7-cm diameter) down to 90 cm at the same location as the monitoring stations (six in total) and then cut in 0–25, 25–50 and 50–90 cm layers. Bulk density (BD) was estimated by the core method [58], and particle size distribution through laser diffraction (Mastersizer 2000, Malvern Instruments). A dedicated algorithm was used to convert diffraction values into pipette values [59].

In addition, soil texture retrieved from the previous study by Longo et al. [60] was used to analyze EC-texture correlations across the fields (Figure 1).

## 2.3. Electrical Resistivity Tomography

Surveys were collected using a Syscal Junior 72 resistivimeter (Iris Instruments, Orleans, France) with a Wenner array on a transect line of 14.1 m composed of 48 stainless-steel electrodes spaced 0.3 m. Electrodes were hammered into the first few centimeters of the ground, looking for the best compromise to ensure electrical contact and point-like current injection. The measured contact resistances had values ranging from 0.1 to 5 kΩ, and the stacking quality factor "Q" was set to 1% (3 to 6 stacks). In this study case, first, all apparent resistivities $\rho_a$ outside the range 0–30 Ωm (about ten rejected data in total) were deleted, and then the inversion process assuming a 5% error model was performed. The inversion process of the acquired dataset has been performed with the ResIPy software [61], based on the R2/R3t codes based on Occam's inversion method [21].

To process time-lapse surveys, the software takes the first dataset as background model and then inverts for the differences between consecutive time steps [61,62]. In this way, we obtained changes in EC (difference %) during time by subtracting the first reference survey from the following ones, converging in few iterations to a final RMS close to 1.

## 2.4. Frequency-Domain Electromagnetic Method

The EMI (based on a frequency-domain—FDEM—approach) surveys were performed on the same dates as the ERT acquisitions in order to map the apparent EC (EC$_a$) of the subsurface. The electro-magnetometry technique in the frequency domain utilizes low-frequency (~1–100 kHz) time variations in electromagnetic fields that originate at or near the surface and diffuse into the subsurface, measuring the interaction between an induced primary electromagnetic field and the resultant secondary electromagnetic field. In particular, the operating principle of the terrain conductivity meter is based on classical EM induction theory [38,63]. As schematically shown in Figure 2, a time-harmonic current (with frequency in the order of the kHz) passes through the transmitter loop (Tx), and the primary magnetic field $H_p(t)$ generated in the transmitter is in-phase with the current. A conductive ground responds to the time-varying primary magnetic flux by establishing a system of electromagnetic eddy currents, whose secondary magnetic field $H_s(t)$ tends to oppose the change $\partial H_p/\partial t$ of the primary flux. The secondary magnetic field $H_s$ is the phase-shifted from $H_p$ as a result of the induced currents. The magnitude of the phase shift with the primary magnetic flux depends on the electrical conductivity distribution of the ground. In general, the secondary magnetic field is delayed (and much smaller than) with respect to the primary magnetic field but still detectable as its phase shift is close to $\pi/2$ with respect to the primary field.

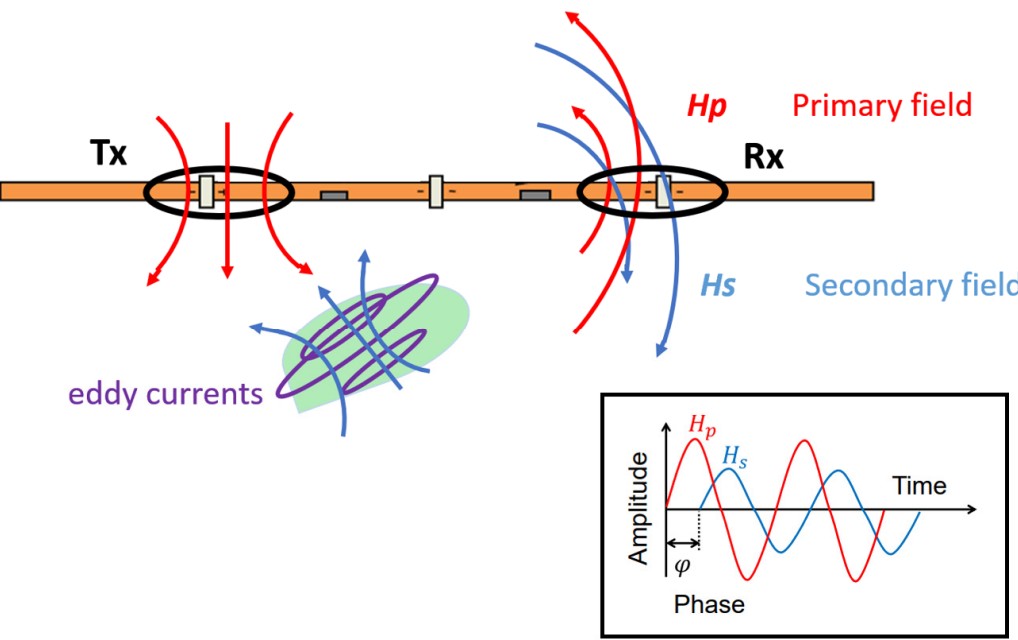

**Figure 2.** Schematic representation of a frequency-domain electromagnetic (FDEM) device, with the corresponding primary and secondary magnetic fields (Hp-Hs) amplitudes and phase lags.

When measuring, both primary and secondary electromagnetic fields are sensed by the receiver coils, and from their ratio, a depth-weighted $EC_a$ can be derived. The ratio $H_s/H_p$ is a complex number composed of an in-phase component and an out-of-phase, or quadrature, component. This ratio is related to the instrument characteristics (operating frequency, coil separation, and orientation) but also to the ground properties (magnetic, conductive, and dielectric). Typically, dielectric properties can be ignored in the kHz frequency range, and considering that most of the subsoils are practically non-magnetic, the magnetic permeability μ of the ground is assumed to be equal to that of free space ($\mu_0 = 1.257 \times 152\ 10^{-8}$ H/m) [38]. Therefore, when the so-called "Low Induction Number" (LIN) conditions are verified (i.e., when the induction number β is <<1 [64]):

$$\beta = s\sqrt{\frac{2}{\omega\mu_0\sigma}} \tag{1}$$

[σ is the conductivity of the soil, ω is the angular frequency (ω = 2πf) of the signal, and s is the separation of the two coils] it is possible to derive the apparent conductivity $\sigma_a$ of the ground, assumed as a homogeneous half-space, as:

$$\sigma_a = \frac{4}{\mu_0\omega r^2}ImQ \tag{2}$$

where *ImQ* is the imaginary ratio of secondary-to-primary field (Quadrature).

In our case, EMI data were collected using the GF Instruments CMD-Mini Explorer (GF Instruments, Brno, Czech Republic), which operates at 30 kHz with a combination of three coil spacing (0.32 m, 0.71 m, 1.18 m). Since the focus of this study was the shallowest portion of the soil (<1 m), only the Vertical Coplanar Orientation (VCP) mode that is more sensitive to the shallow subsurface, with nominal exploration depths of 0.25, 0.5, and 0.9 m, was acquired and examined.

For each survey, the device was carried at the soil surface, placed on a dedicated wood sledge, pulled by a tractor, and linked to it by a 4-m long rope. The travel speed was approximately 7 km h$^{-1}$, and the parallel transects were set about 7 m apart from each other. Measurements were logged every 0.5 s and paired with coordinates obtained from ProXT GPS receiver (Trimble, Sunnyvale, CA, USA) with decimetric accuracy.

On each survey date, an average of 5000 points were acquired for each monitored field. The surveys were carried out five times and distributed over a five months period, for a total of ten data sets (2 treatments—CONV or CONS—by 5 surveys each). Since rainfall induces sudden increases in soil moisture, at least in shallowest soil layers, an attempt was made to conduct field measurements after significant rainfall events.

The measured data were filtered from outliers (values outside the mean $\pm$ 3 standard deviations) and applying a smoothing window, replacing each data point with the average of its neighbors (size = 5). Since EC is also sensitive to temperature, a correction is needed for a proper interpretation of the surveys [65]. In this study, $EC_a$ values were corrected using the model proposed by [35]:

$$EC_{25} = EC_T \times 0.4470 + 1.4034^{-\left(\frac{T}{26.815}\right)} \tag{3}$$

where $EC_{25}$ (mS/m) is the apparent electrical conductivity at 25 °C and T is the temperature (°C) at the time of measurement. Note that the $\sigma_a$ measured in the EMI surveys are "apparent" since they represent integrated values over depth. Inverse methods need to be used to convert the $EC_a$ measurements to a depth profile of EC [37,38]. The datasets were inverted with the code EMagPy [38], using the Cumulative Sensitivity (CS) forward model and the L-BFGS-B (Broyden–Fletcher–Goldfarb–Shanno) optimization method [66] to minimize the total misfit between observed values and predicted values from the forward model solution. Note that under the CS simplified assumption, three measurement configurations, as in our case, are sufficient to produce an inverted profile as a function of depth. EMagPy has the capability to perform quasi-2D inversions, generating inverted EC depth profiles for each point of measurement. Four layers were set in the initial inversion model, with interfaces at 0.25, 0.50, 0.90 m and >0.90 m depth. The convergence was achieved for a final RMS misfit close to 1 for each survey.

### 2.5. Statistical Analysis

Treatment (CONV vs. CONS) effects on BD at the end of the experiment were tested through a one-way ANOVA. The regression model between EC and VWC, soil temperature, BD, sand, and clay was estimated by multiple stepwise regression with backward selection on samples collected close to the monitoring stations (5 dates $\times$ 2 fields $\times$ 3 stations $\times$ 3 depths). The significance entry level for an independent variable was set as $p < 0.05$. The possible correlation between field variability of EC and textural classes obtained in the 18 positions per field was also analyzed by estimating Person's coefficients. The same statistical method was applied between VWC and soil physical properties (BD, sand, silt, and clay) at the three monitoring stations. The statistical analysis was performed using Statistica (StatSoft Inc., Tulsa, OK, USA) and R (R Core Team, Vienna, Austria).

## 3. Results

### 3.1. Weather and Soil Monitoring

Throughout the experiment, the largest rainfall events were recorded in February and March (96 and 113 mm, respectively), while minor events (2 to 5 mm) occurred in the first months. Rainfall cumulated in a total of 266 mm with a maximum event of around 17 mm, prior to the third survey (Figure 3).

Except for a drop in mid-March, the average air temperature remained over zero in the winter season (4.5 °C, on average), fluctuating between $-6.8$ °C and 15.7 °C. In April, the temperature started rising rapidly, reaching its maximum (27.3 °C) on 20 April 2018. Shallow subsurface temperature (<0.1 m depth) varied between CONS and CONV, respectively, averaging 5.1 °C and 5.2 °C in the colder months and 14.6 °C and 13.1 °C in the warmer months. CONV was also characterized by slightly higher daily fluctuations, considering the standard deviation equal to 4.8 °C, compared to 3.8 °C in CONS. Variations decreased in the deeper layers, where values between the treatments were comparable, 2.6 °C on average.

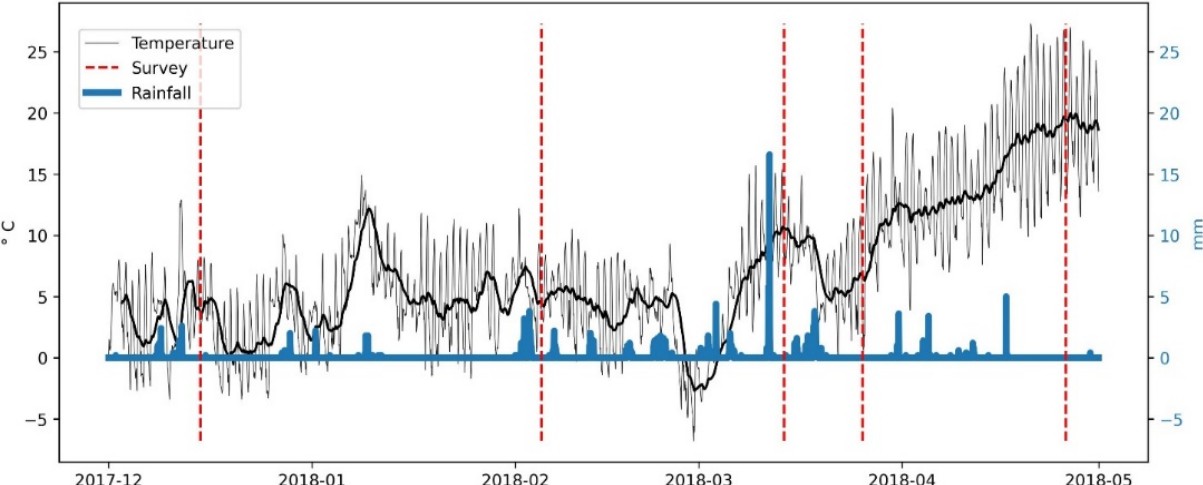

**Figure 3.** Time series of rainfalls [mm] and air temperatures [°C] from the weather station (December 2017–May 2018). The solid black line shows daily mean temperature values. The dashed red lines show the time of ERT and EMI surveys.

The soil of the three monitoring stations was classified as silt loam. Soil VWC also varied according to the treatment (CONV or CONS—Figure 4). In general, daily values of CONS appeared more homogenous along the soil profile, while larger variations were observed in CONV (variation coefficient: 0.05 vs. 0.08). For instance, lower values were recorded in the shallow subsurface in CONV (ranging from 26.2–39.3%) compared to deeper layers (25.2–44.9%). Considering the middle layer, the two treatments were comparable in the colder months, while CONS dropped down to 21.8% from April. At 55 cm, CONS remained stable at 37.5% during dry periods while settling around 45% after rainfall events. Conversely, CONV values had lower variations after rainfall events averaging 38.4% from December to March and 33.9% from March to the end of the experiment. Similar behavior was observed at the intermediate monitoring depth (35 cm).

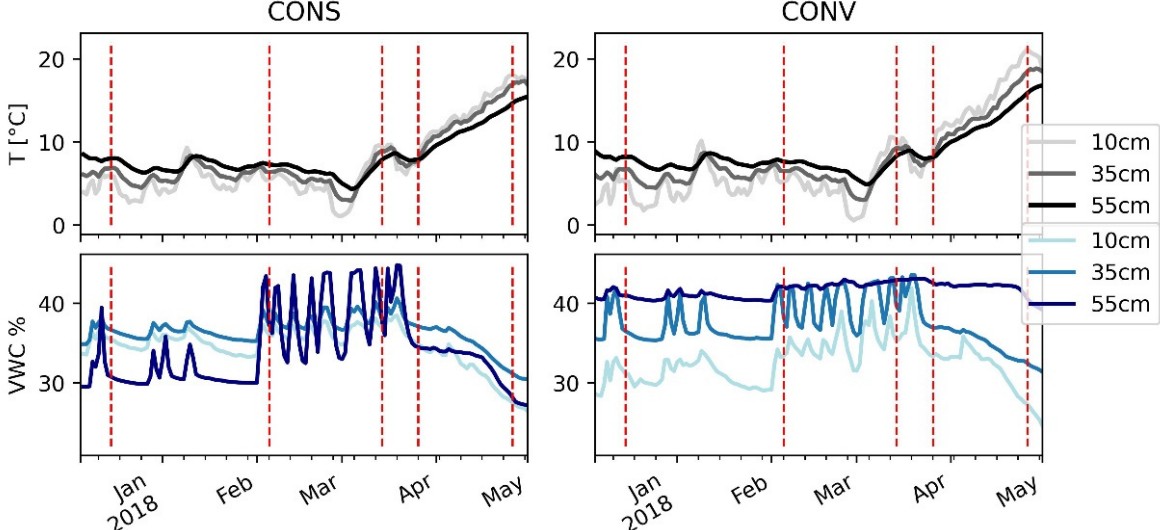

**Figure 4.** Time series of soil temperature [°C] and soil volumetric water content [%] from the sensors installed in the two field sites (CONV and CONS). Dashed red lines show the time of ERT and EMI surveys. Data shown are the average of the three stations for each treatment.

Soil BD was significantly affected by the different treatments in all three layers, being 1.28 ($\pm$0.08) g cm$^{-3}$ vs. 1.08 ($\pm$0.13) g cm$^{-3}$ in 0–25 cm, 1.48 ($\pm$0.08) g cm$^{-3}$ vs. 1.39 ($\pm$0.09) g cm$^{-3}$ in 25–50 cm, and 1.51 ($\pm$0.02) g cm$^{-3}$ vs. 1.34 ($\pm$0.14) g cm$^{-3}$ in

50–90 cm in CONS and CONV, respectively. In general, BD positively affected VWC in the top 25 cm (r = 0.464, *p* < 0.05) (Table 2). On the other hand, the texture influence prevailed in the deeper layer, being negatively correlated with sand (r = −0.688) and positively correlated with silt and clay (r = 0.703 and 0.552, respectively).

**Table 2.** Correlation between VWC and BD, Sand, Silt, and Clay at different soil layers. Significant (*p* < 0.05) Pearson's *r* is reported in bold.

| Depth (cm) | BD | Sand | Silt | Clay |
|---|---|---|---|---|
| VWC (0–25) | **0.464** | 0.358 | −0.372 | −0.267 |
| VWC (25–50) | 0.213 | 0.310 | **−0.398** | 0.078 |
| VWC (50–90) | −0.297 | **−0.688** | **0.703** | **0.552** |

*3.2. Electrical Resistivity Tomography*

Figure 5 shows the variation of the inverted electrical resistivity (hereafter ER) 2D sections over the course of the experiment for the CONS and CONV treatments. Note that ER reached a maximum value of 100 Ωm, although the figure scale is limited to 25 Ωm.

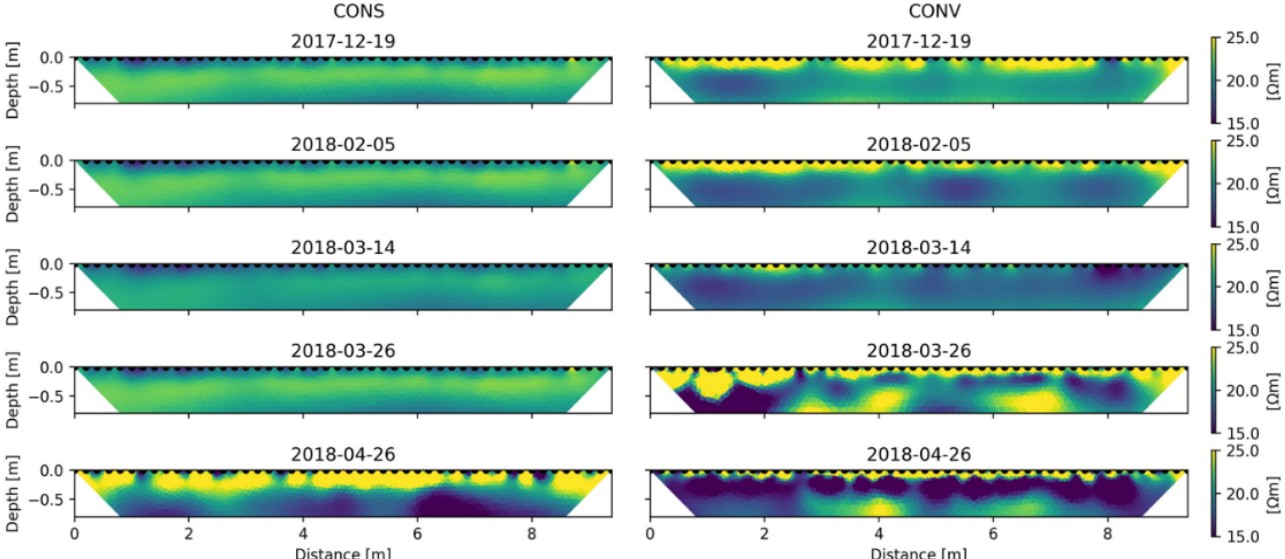

**Figure 5.** Absolute inverted ER values along the ERT transects over time, respectively, for the conservative (**left**) and the conventional treatment (**right**).

For the CONS, the spatial distribution of ER did not vary from December 2017 to March 2018. The very shallow subsurface (down to about 0.25 m depth) was more conductive with respect to the deeper subsurface. In April 2018, contrasts of ER were enhanced by the presence of an intermediate resistive layer (20 Ωm, on average) ranging from 0.25 to 0.5 m depth. Note that for the CONS treatment, there was no appreciable lateral variability, while a clear variation is observed with depth.

Conversely, for the CONV treatment, the ER distribution is considerably more heterogeneous in space and in time. In this case, the shallowest layer was resistive as compared to the underlying layers for all time steps except for the mid-March survey. The contrasts of ER between the layers were higher in the case of the CONV treatment as compared to the CONS treatment.

In Figure 6, the same data were inverted using a time-lapse inversion approach and revealed variations in both treatments that occurred on 14 March as a result of an important rainfall event (see Figure 3) with a decrease of the ER up to 10% homogeneously all over the section in the CONS treatment. At the same time, the same trend was observed for the CONV treatment, but with a higher decrease of ER (up to 20%) in the shallower layer. On

March 26th, ER did not change (as compared to the background time) for the CONV, while a strongly scattered increase and decrease of ER were observed in the CONV treatment. Finally, the last time step (April 2018) showed an increase of ER (up to 25%) in the shallower layer both in CONS and CONV, but involving different depths, i.e., the top 25 cm in the CONS and 10 cm in the CONS treatment, associated with a decrease in the ER underneath (−20%).

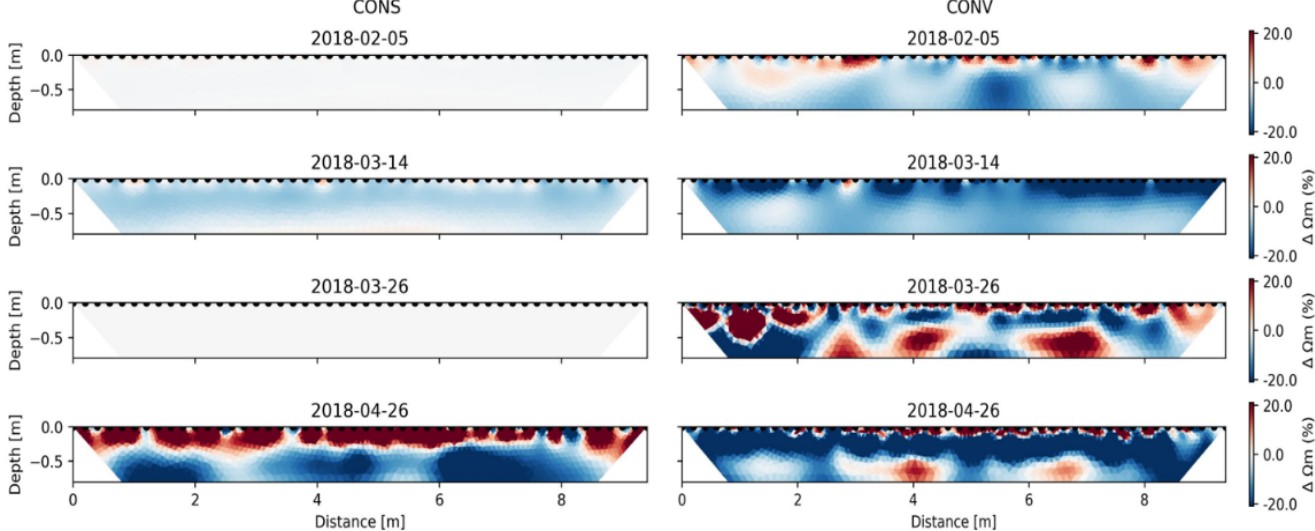

**Figure 6.** Comparison of time-lapse ER differences between the background time (19 December 2017) and the following times, respectively, for the conservative (**left**) and the conventional treatment (**right**).

### 3.3. Frequency-Domain Electromagnetic Method

The inverted EC from EMI surveys ranged from 1.9 to 92.1 mS m$^{-1}$, with increasing values with depth. The spatial patterns of higher/lower values were similar to what was observed in ERT surveys across depth and time. At the time of the first survey, the average topsoil values of CONV were lower than in CONS (19.7 mS m$^{-1}$ vs. 29.3 mS m$^{-1}$), while differences decreased in the middle layer and were comparable with each other in the deeper (50–90 cm) layer (44.3 mS/m, on average) (Figure 7). As for ERT, values did not vary much across the first four dates (coeff. var. = 0.41) (Figure 8), despite being the February 2nd conductivity slightly lower on the surface and higher in the deep layer of CONS, which had the highest conductivities for the entire study. Conversely, the last survey showed low EC in CONS (decreasing by 15% with respect to the initial conditions—Figure 8), averaging 6.3 mS m$^{-1}$ against a value of 21.0 mS m$^{-1}$ in CONV in the 0–25 cm soil layer.

### 3.4. Comparison between EC and Soil Properties

Soil VWC, BD, and sand content significantly influenced EC according to the multiple linear regression results. A negative correlation was observed with sand ($\beta_2 = -0.32$), while VWC and BD positively affected EC ($\beta_2 = 0.434$ and $\beta_2 = 0.366$, respectively).

Considering the 18 soil samples collected in the whole fields (Figure 1), EC values were spatially influenced by texture in both treatments (Figure 9), being significantly ($p < 0.05$) positively correlated with clay and silt content (correlation coefficients were 0.21 and 0.22, respectively) and negatively correlated with sand (−0.22). Correlations were stronger in CONS, reaching Person's coefficients up to 0.86 for clay. Moreover, texture-EC correlations were higher in the 25–50 cm layer while decreased in 50–90 cm, where a higher sand content was observed. For the same textural class, the EC values differed between the CONS and CONV treatments, especially at the first layer (0–25 cm). For instance, for a silty clay loam type, CONV-EC was about 25 mS m$^{-1}$, while for the CONS-EC was about 45 mS m$^{-1}$. Differences between CONV and CONS tended to decrease with depth, as expected.

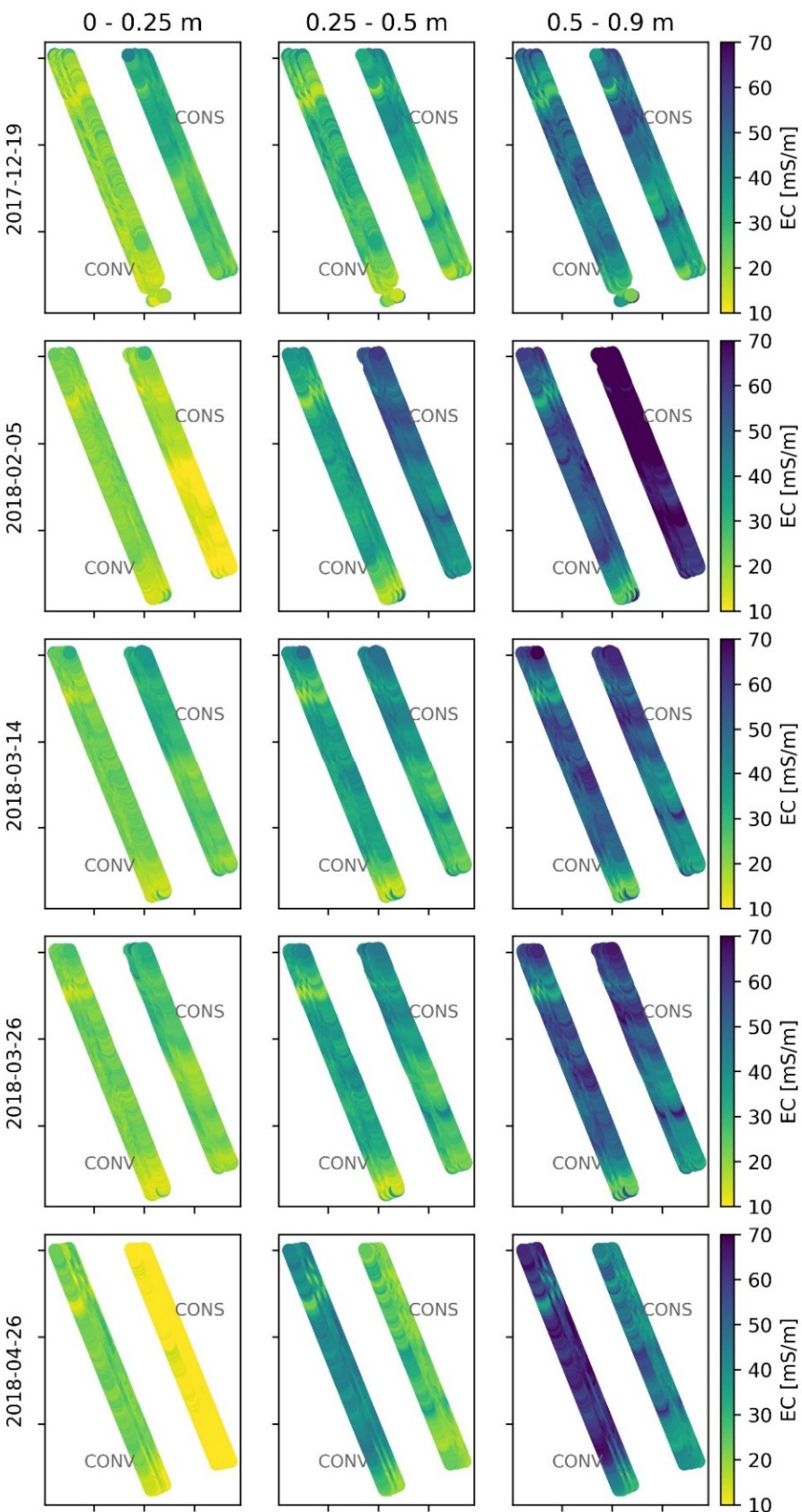

**Figure 7.** Comparison of EMI response over time between the two treatments. The three columns correspond to the three layers (plus infinite subspace below) used for the EMI inversion.

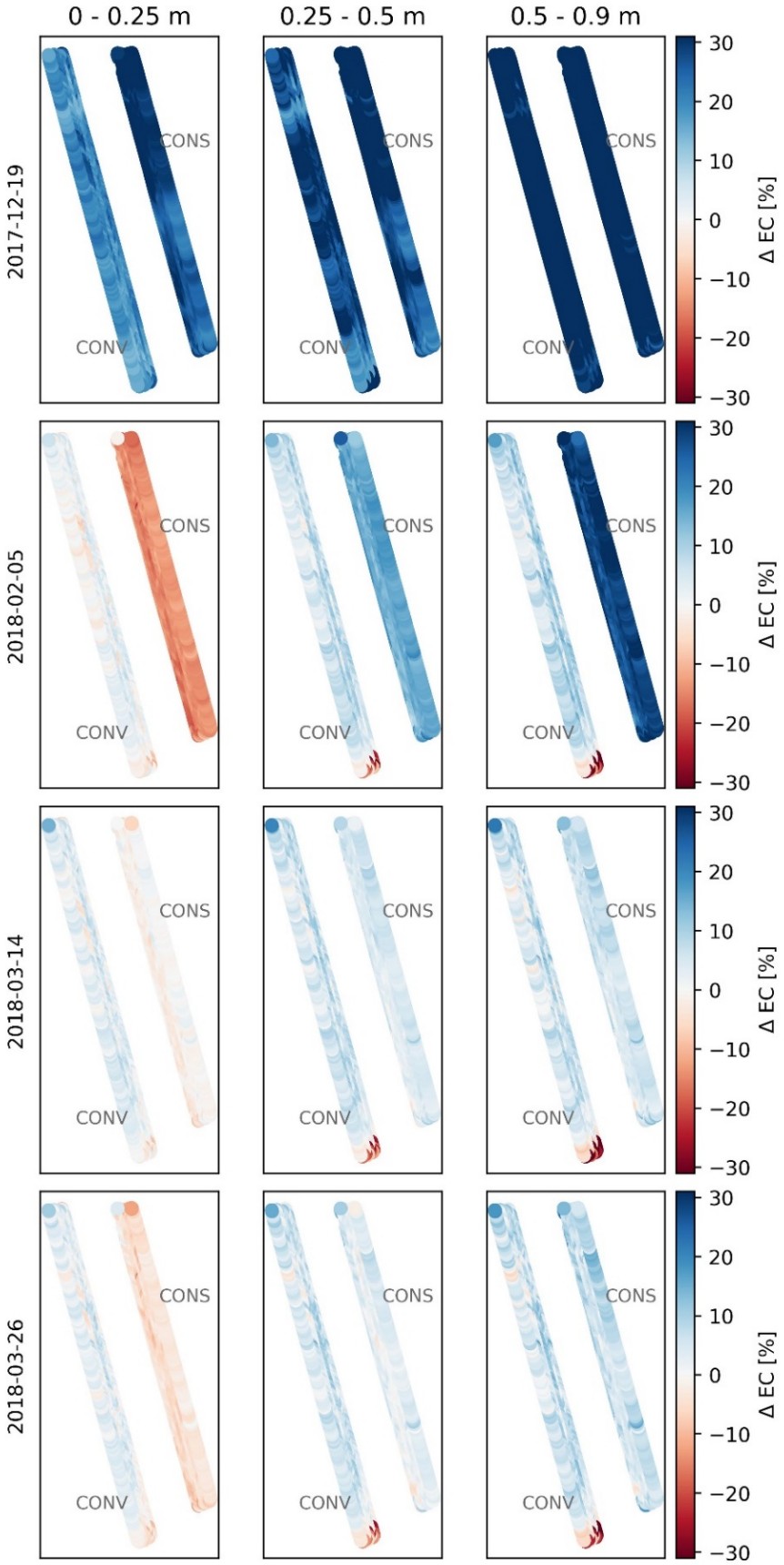

**Figure 8.** EMI inversion results: comparison of time-lapse EC differences between the two treatments.

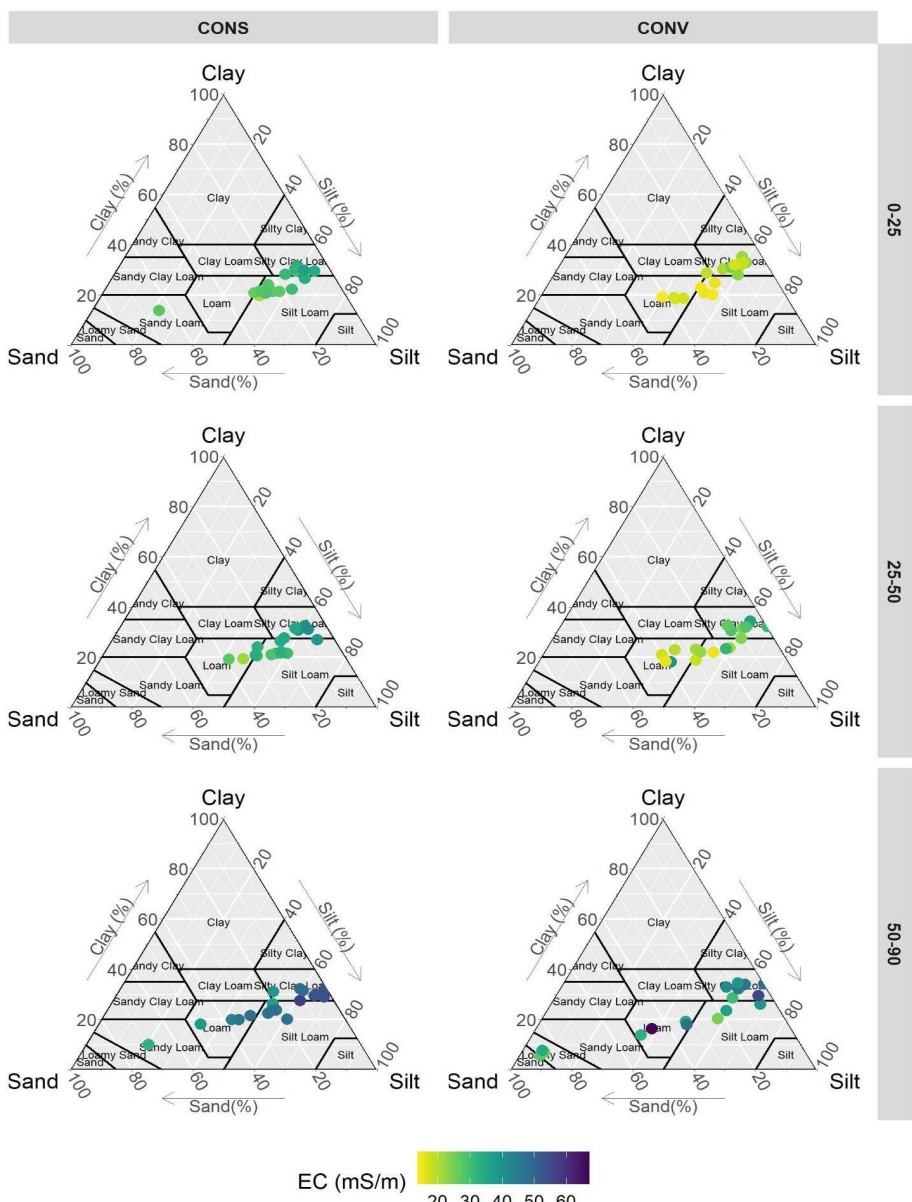

**Figure 9.** Relations between textural classes and inverted EC measured on 19 December 2017.

## 4. Discussion

Soil covering has a crucial effect on VWC [67,68]. On vegetated soil, transpiration dominates the evapotranspiration process being proportional to crop biomass and root growth and the water input [69,70]. Moreover, different root architectures may affect water uptake [71]. On the opposite, transpiration is considered null on bare soil. In this study, the soil was bare on CONV while vegetated on CONS, influencing VWC dynamics [72–74]. The effects of soil properties on VWC varied along the soil profile. Indeed, the soil BD was found as the dominant variable affecting VWC above 25 cm, presumably due to the different treatment (i.e., tillage) effects on soil structure and, in turn, on water dynamics. In fact, the CONV BD was lower at topsoil due to tillage operations [75]. On the contrary, the texture effect was evidenced below 50 cm revealing how deeper soil horizon dynamics might primarily be affected by native soil structure, which was possibly maintained in CONV while appearing more compacted in CONS. It can be speculated that repeated passages of heavy agricultural machinery (e.g., combine and direct driller) might have caused subsoil compaction in this soil that was previously demonstrated to be prone to soil compaction in the absence of tillage [11,76]. Nevertheless, the quick response of deep

VWC at rain events, particularly from February during the rapid vegetative growth of the winter wheat in CONS, might prove the presence of functional pores. Indeed, as already evidenced in the same site by Camarotto et al. [57], the rapid load and discharge of the water-filled pore space might suggest the presence of a few vertical macropores that locally improved the hydraulic conductivity (data not shown) [77,78]. This is also supported by the similar dynamics observed in time between the two treatments, involving however different layers. In CONV, VWC was particularly responsive to rainfall in the 0–35 cm and not in the deeper one. On the contrary, in CONS, the increase in the 0–35 cm was less pronounced, while peaks were observed in the 55-cm layer, suggesting rapid infiltration movements from surface layers. These dynamics were observed at all three replications (i.e., monitoring stations) for field. Moreover, results from the previous two years showed similar fluctuations between tilled soils with cover cropping and CONV, but not with CONS [57], suggesting an effect given by the absence of tillage combined with cover crop root apparatus [52,71]. Furthermore, a different influence of the shallow water table was excluded being the experiment fields close one another (distance of 60 m).

The VWC dynamics are usually also affected by soil texture. The monitoring stations had a similar texture (i.e., silt-loam), and therefore, it was not possible to directly assess the effect of different textures on soil water dynamics. However, previous studies have already demonstrated how sandy soils have a limited ability to retain water because the pore size distribution consists mainly of large pores. Conversely, highly-porous fine-textured soils strongly retain water within micropores [79,80].

Based on approximately five months of soil monitoring for the two farming practices, it was possible to notice a significant consistency in the response of the measured parameters between the different sensing methods. The results of the EMI surveys confirmed an expected strong direct correlation between EC and soil water distribution [81,82]. Additionally, time-lapse inversions made it possible to somewhat eliminate the static effects of soil properties (e.g., texture, mineralogy) [83], thus highlighting the dynamic part of the response. Thus, as already reported by Blanchy et al. [84], the measured EC range rose in the different surveys just after rainfall events, and the higher moisture areas were easily identifiable as the most conductive portions both in the ERT and EMI inversions.

The finer resolution of ERT allowed for better observations of depth-specific properties at the expense of a more complex setup (i.e., placing the electrodes and logistics). However, the measurement only referred to local portions of the fields, having been conducted on transects adjacent to the central monitoring stations. ERT models showed more homogeneity for the CONS treatment, resulting in a higher sensitivity to changes in the soil water dynamics. In this case, since the soil structure remained stable over time, it was more difficult to observe changes in ER. At the same time, the continuous structure remodeling of the tilled soil (CONV) increased the heterogeneity of pore architecture [85]. Time-lapse ERT measurements gave clear evidence that the increase in soil moisture following rainfall events produced strong effects. Therefore, the persistent conductive regions within the inverted models are likely due to higher moisture content, being in line with previous studies [11].

Strong heterogeneities in the CONV profile at the beginning of spring may have been associated with two distinct causes: (i) a soil crust, confirmed by visual assessment, generated at the end of March by heavy raining events which followed the drier conditions observed during winter and (ii) the harrowing operations in April, which broke up the soil aggregates and helped dry the first layer [11]. This hypothesis is also supported by the soil moisture gradient observed within the profile (i.e., drier above and wetter below). It is plausible that the soil crust also acted as a barrier, maintaining favorable conditions for plant growth in the deeper layers, as also illustrated in the modeling work of Assouline et al. [86] and in the results of Cassani et al. [12]. Similarly, the harrowing operations could have led to air-filled macroporosity at the topsoil (more resistive layers), contributing to increasing water storage in the deeper layers (less resistive layers). Nevertheless, a similar layering was also observed in the CONS transect, where the increase in ER (>20% variations) at

the topsoil (top 25 cm) can likely be related to crop water uptake. For instance, as recently documented [83], cover crops showed considerably lower EC compared to bare soil, which authors implied was due to lower soil moisture storage, and in this study, reflected on the ER distribution.

BD also affected ER in CONV, showing less dense layers at the very surface and deeper layers and higher in the intermediate layer, which is clearly visible from the ERT models and in agreement with other authors [87]. In contrast, BD in CONS increased with depth, slightly reducing VWC.

On the other hand, it is clear that EC distribution from the EMI inversion process is not directly comparable with the ER tomographic transects. Notably, the clear differentiation between CONS and CONV electrical profiling was slightly lost in the EMI models, strongly dominated by the inherent heterogeneity of the areal survey [22,42]. Indeed, EC values were significantly affected by field texture and bulk density which strongly contribute to electrical properties and hydrological dynamics [22,88]. Note that the lower depth resolution of EMI with respect to ERT has a clear impact on these results.

From a structural point of view, the different EMI sections at depth show similar patterns over the acquisition period. Time-lapse analysis revealed that farming treatments influenced the soil moisture dynamics, even with a magnitude of only a few mS m$^{-1}$. From December to April, soil water content was high due to some rainfall events and low evapotranspiration (ET). Conversely, between April and May, the increased ET controlled the soil drying at the beginning of Spring, as widely recognized [89,90].

Fluctuations of shallow EC were greater in CONS than in CONV, as mirrored by the water content variability in the first 10 cm layer. As already observed in the ERT models, this can be reasonably associated with the increased crop water uptake of the cover crop in CONS at the start of Spring [90].

Considering the deepest portion of the models (0.5–0.9 m depth), the variability was always higher in CONV than in CONS, with a coefficient of variation of 0.15–0.21 and 0.11–0.14, respectively. Most likely, the electrical behavior was affected by deep infiltration increased by the saturated hydraulic conductivity (data not shown), two orders of magnitude higher in CONV than in CONS, as already reported, e.g., by other authors [87,91].

It should be noted that all EMI inverted models have low depth accuracy since they are generated by a pseudo-2D process from a depth-weighted measurement [38]. In this respect, the penetration depth was lower than those predicted by the instrument manufacturer. We calculated, for each coil configuration, a sensitivity profile of the measurements related to the depth. The inverted EMI models presented here were limited to the depths where the normalized sensitivity of the measurements reaches zero, approximately at 0.2, 0.4, and 0.7 ratios of coils separation. Moreover, if compared to DC resistivity techniques, EMI data are more sensitive to distortions since the propagation of the electromagnetic field depends on more physical properties of the ground (i.e., magnetic, conductive, and dielectric properties). Hence, EMI measurements have the great advantage of being quick and contactless, but they are prone to more uncertainties related to a number of issues (i.e., electronic instrumental drift, heterogeneous height and orientation of the device, air temperature variations during the acquisition, variations of ground surface cover [27]).

## 5. Conclusions

This study supported the strong effects of agronomic management systems on soil properties, such as bulk density and water content. The direct comparison between conventional and conservation treatments in a five-month period confirmed that soil electrical behavior can be a relevant proxy of the soil structure changes. The ERT survey highlighted a strong variability of tilled soil, driven by structure remodeling, while the EMI results better emphasized topsoil EC variations. Furthermore, our findings substantiate the importance of coupling geophysical surveys with direct method assessments (e.g., soil samplings) for a complete understanding of the soil processes.

In conclusion, this study paves the way for advancing knowledge of soil dynamics by identifying key soil parameters that can capture spatial and temporal changes in soil.

**Author Contributions:** Conceptualization, F.M. and A.C.; methodology, A.C., M.L., I.P. and F.M.; validation, G.C.; investigation, M.L. and I.P.; data curation, A.C. and M.L.; writing—original draft preparation, A.C.; writing—review and editing, A.C., B.M., M.L., I.P., G.C. and F.M.; funding acquisition, F.M. All authors have read and agreed to the published version of the manuscript.

**Funding:** The research leading to these results was in part funded by the European Union under grant agreement LIFE12 ENV/IT/000578 (LIFE HELPSOIL project) and in part by the European Union and the MiPAAF under ICT-AGRI-FOOD COFUND 2019 ID 40922 (SoCoRisk project).

**Data Availability Statement:** The data supporting the conclusions of this article will be made available by the authors on demand.

**Acknowledgments:** The authors are grateful to Felice Sartori for their precious support during the investigation operations. The authors also thank Veneto Agricoltura for the availability of the test fields and the staff of the experimental farm "L. Toniolo" of the University of Padova for the operational support. BM acknowledges the financial support from European Union's Horizon 2020 research and innovation program under a Marie Sklodowska-Curie grant agreement (grant no. 842922).

**Conflicts of Interest:** The authors declare no conflict of interest.

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
