# Peer review of "Electro-Magnetic Geophysical Dynamics under Conservation and Conventional Farming"

_remotesensing, doi:10.3390/rs14246243_

Round 1
Reviewer 1 Report
This study assessed the ability of Electrical Resistivity Tomography (ERT) and Electro Magnetic Induction (EMI) methods for monitoring the effects of conventional (CONV) and conservation (CONS) agricultural practices and analyzed soil water distribution caused by both short- and long-term effects under the two different practices. ER provides a 2D vertical and horizontal electrical image of the subsurface, while the EMI provides only a horizontal electrical image of the subsurface, which both provided sufficient information to distinguish between the effects of CONV and CONS. The authors have done a lot of experimental analysis, but there are still some factors lacking of consideration.
1) More analysis is needed for identifying the influence of climate, i.e., the infiltration effect of precipitation and evaporation effect will affect soil water under different soil texture condition, which should be further discussed.
2) The influence of the initial soil water of the two sites (CONV and CONS) should be considered in the experiments.
3) Different growth stages of crops and different crop species probably impact soil water as plant roots uptake water and precipitation can be intercepted by plants.
4) The author only provided a quantitative analysis about the influence factors. Can a quantitative index of how each factor affects soil water be given to readers for a more intuitive presentation?
Author Response
Comment |
Reply |
This study assessed the ability of Electrical Resistivity Tomography (ERT) and Electro Magnetic Induction (EMI) methods for monitoring the effects of conventional (CONV) and conservation (CONS) agricultural practices and analyzed soil water distribution caused by both short- and long-term effects under the two different practices. ER provides a 2D vertical and horizontal electrical image of the subsurface, while the EMI provides only a horizontal electrical image of the subsurface, which both provided sufficient information to distinguish between the effects of CONV and CONS. The authors have done a lot of experimental analysis, but there are still some factors lacking of consideration. |
We thank the Reviewer for pointing out these issues that were promptly considered for manuscript improvement. |
1) More analysis is needed for identifying the influence of climate, i.e., the infiltration effect of precipitation and evaporation will affect soil water under different soil texture condition, which should be further discussed.
|
We got the point of the Reviewer, and we agree. Actually, the two fields are a few tens of meters apart, so they are not affected by a different meterological conditions and are also quite similar in terms of texture. Indeed i) they present the same texture gradient across the entire field surface (Figure 9) and, ii) the soil of the three measuring stations falls in silt loam class. Therefore, it was not possible to assess the effect of different textural classes on soil water content. For more clarity these findings are now reported in L315 and LL441-444. According to the Reviewer’s comment, more details on the effect of treatments on evapotranspiration process and, in turn, on VWC is now described in LL420-422. |
2) The influence of the initial soil water of the two sites (CONV and CONS) should be considered in the experiments.
|
We thank the Reviewer for the suggestion. We wish to highlight that this experiment was running from 2011, therefore the initial soil conditions are known and were homogeneous between the two sites. Therefore, the differences in soil water content observed in this study reflected the effects of agronomic managements. We now included this information in the Materials and Methods (LL131 and 140-141). |
3) Different growth stages of crops and different crop species probably impact soil water as plant roots uptake water and precipitation can be intercepted by plants.
|
We thank the Reviewer for the suggestion. The text has been revised accordingly in LL161-162, L420-424. |
4) The author only provided a quantitative analysis about the influence factors. Can a quantitative index of how each factor affects soil water be given to readers for a more intuitive presentation? |
Thanks for the suggestion. A new paragraph has been introduced reporting the correlation between VWC and the soil properties (LL295-298) which is then further discussed in the discussion (LL329-332). |
Reviewer 2 Report
Brief summary
The use of geophysical tools (ERT and EMI) to analyse the impacts of various management approaches (conventional and conservative) on the soil is evaluated in this research, particularly with regard to soil water content and water redistribution along the soil depth.
Broad comments
In general, the work is of a very good standard.
Regarding how the geophysical data were obtained and how the inversion process worked, the information supplied is very accurate.
On the other hand, there are certain aspects of the variability of the soil properties that might be clarified. In particular, the two experimental fields (CONS and CONV) appear to exhibit some interfield and intrafield variability, which could affect the inferences made from the experiments.
Specific comments
L 137 - Hypocalcic Calcisols - The final 's' is used for the plural: speaking of a specific soil, it must not be used: Hypocalcic Calcisol.
L 149-150 – Are you sure that the two fields do not differ? Isn't there variability within the field? In Fig. 9, the texture classes are loamy sand, sandy loam, silt loam, silty clay loam...
L 165 – Mean values without standard deviation? How many sampling points?
L 165 – Assimilable P – Not “assimilable” but “available”.
L 187-188 – I guess the sampling was done at 10, 30 and 55 cm: however, it should be specified.
L 189 – The bulk density was obtained with the core method: but what is the volume of the core sampler?
L 330 – In Fig. 4, the temperature and water content data refer to a single measuring station for CONS and CONV; but there are three stations per field. Why not report average values?
L 391-393 – The bulk density values for the three layers do not show the standard deviation.
L 434 – The heterogeneity of the soil profile is considered to have been impacted by a soil crust in CONV. But there is no evidence of the effective presence of a surface crust, which is not difficult to observe in the field.
L 472-474 – Ksat is higher in CONV than in CONS, probably due to the lower bulk density caused by ploughing. But BD is also markedly lower in depth, beyond the lower limit of the tilled horizon: this finding would deserve comment.
Author Response
Brief summary
The use of geophysical tools (ERT and EMI) to analyse the impacts of various management approaches (conventional and conservative) on the soil is evaluated in this research, particularly with regard to soil water content and water redistribution along the soil depth.
Broad comments
Comment |
Reply |
In general, the work is of a very good standard. Regarding how the geophysical data were obtained and how the inversion process worked, the information supplied is very accurate. On the other hand, there are certain aspects of the variability of the soil properties that might be clarified. In particular, the two experimental fields (CONS and CONV) appear to exhibit some interfield and intrafield variability, which could affect the inferences made from the experiments.
|
We thank the anonymous Reviewer for his/her constructive comments which were all addressed. Please see the replies below. |
Specific comments
L 137 - Hypocalcic Calcisols - The final 's' is used for the plural: speaking of a specific soil, it must not be used: Hypocalcic Calcisol. |
Revised, thank you. |
L 149-150 – Are you sure that the two fields do not differ? Isn't there variability within the field? In Fig. 9, the texture classes are loamy sand, sandy loam, silt loam, silty clay loam...
|
The two fields showed the same textural trends within the ca. 1 ha area each (see Figure 9). However, the monitoring stations located in three positions per field was characterized by the same textural class (silt loam). These details have now been added to clarify this issue (L315 and L441). |
L 165 – Mean values without standard deviation? How many sampling points?
|
Thanks for the comment. These are the main soil physico-chemical properties measured on composite soil sample at the beginning of the experiment (2010). |
L 165 – Assimilable P – Not “assimilable” but “available”.
|
Changed in L164 |
L 187-188 – I guess the sampling was done at 10, 30 and 55 cm: however, it should be specified.
|
The sampling has been performed by continuous undisturbed coring down to 90cm and cut at 0-25, 25-50 and 50-90 cm layers. These details have been added in L186 - 188. |
L 189 – The bulk density was obtained with the core method: but what is the volume of the core sampler? |
Core dimension are now added in LL186. |
L 330 – In Fig. 4, the temperature and water content data refer to a single measuring station for CONS and CONV; but there are three stations per field. Why not report average values?
|
Thanks for the comment. Actually, the reported data in Fig. 4 are the average values from the three stations. We have added the information in the Figure caption L343 for more clarity. |
L 391-393 – The bulk density values for the three layers do not show the standard deviation. |
The standard deviations have been added (L327 - 328). |
L 434 – The heterogeneity of the soil profile is considered to have been impacted by a soil crust in CONV. But there is no evidence of the effective presence of a surface crust, which is not difficult to observe in the field.
|
We got the point of the Reviewer, and we agree. We understand that it is difficult to have evidence of the effective presence of soil crust. However, soil crust was verified with visual assessment. Indeed, the high silt and low SOC content soil is prone to soil crust formation.
A and a new sentence clarifying this finding has been added in L470 - 471. |
L 472-474 – Ksat is higher in CONV than in CONS, probably due to the lower bulk density caused by ploughing. But BD is also markedly lower in depth, beyond the lower limit of the tilled horizon: this finding would deserve comment. |
We thank the Reviewer for the comment. The text has been revised including also this aspect in L434-441. |
Reviewer 3 Report
The target of this manuscript is the investigation of the ability of Electrical Resistivity Tomography (ERT) and Electro Magnetic Induction (EMI) methods for monitoring the effects of conventional (CONV) and conservation (CONS) agricultural practices. The study is very well written and easy to understand. However I have some important concerns as regards its suitability for publication in remote sensing journal.
My first important concern is related to the relevance of the manuscript with Remote Sensing journal. The target of the study and the methods used are not related with remote sensing journal. Electro Magnetic Induction and Electrical Resistivity Tomography methods are clearly ground-based approaches and especially ERT involves direct measurement of the respective parameters. All other aspects of the study are also not related to Remote Sensing. I believe that the manuscript would be a perfect fit for an Agricultural, Soil water, or either sensors measurements related journal.
Apart from the above I have the following comments.
1. The title is misleading as the hydrological impacts only indirectly and very superficially were investigated.
2. The graph for soil moisture for CONS (on the left) in Figure 4 is very strange. Specifically, I cannot understand how the deeper layer's soil moisture can fluctuate that much and at the same time the upper layers to have almost stable soil moisture values. It seems like the water is coming and leaving from bellow. Especially the rapid drops of soil moisture in the deeper layer while the top layers have steadily high values is very strange. Can you please check? Is there a huge difference in soil properties of the soil layers? This should be investigated further.
3. I feel like the obtained results and their analysis were not conclusive but provided mostly indications that could fit with various interpretations. Parts of the results are interesting and some of the findings seem to be in line with previous studies findings, however, the added value of the current study isn't clear. The obtained results from the various monitoring methods are analysed separately and there isn't any synthesis. Accordingly the main conclusions of the study are mostly based on indications and assumptions and not on the analysis of the data. The reality is that there is a good amount of data that with better / deeper analysis could obtain more solid conclusions.
Based on the above my recommendation is reject and submit in another more relevant journal.
Author Response
General comments:
Comment |
Reply |
The target of this manuscript is the investigation of the ability of Electrical Resistivity Tomography (ERT) and Electro Magnetic Induction (EMI) methods for monitoring the effects of conventional (CONV) and conservation (CONS) agricultural practices. The study is very well written and easy to understand. However I have some important concerns as regards its suitability for publication in remote sensing journal.
My first important concern is related to the relevance of the manuscript with Remote Sensing journal. The target of the study and the methods used are not related with remote sensing journal. Electro Magnetic Induction and Electrical Resistivity Tomography methods are clearly ground-based approaches and especially ERT involves direct measurement of the respective parameters. All other aspects of the study are also not related to Remote Sensing. I believe that the manuscript would be a perfect fit for an Agricultural, Soil water, or either sensors measurements related journal.
Apart from the above I have the following comments. |
We got the point of the reviewer but we are sorry to disagree. The manuscript has been submitted to Remote Sensing special issue entitled "Near-Surface Geophysics: A Remote Sensing Tool for the Shallow Subsurface" (https://www.mdpi.com/journal/remotesensing/special_issues/near_surface_geophysics_remote_sensing_tool_shallow_subsurface) We sincerely believe that our MS falls within this special issue aims, and, in particular, that it “combined approaches to complex 3D characterization and modeling of surface and subsurface targets and/or processes.” |
Specific comments
1. The title is misleading as the hydrological impacts only indirectly and very superficially were investigated. |
We thank the Reviewer for the suggestion, we revised the title accordingly. |
2. The graph for soil moisture for CONS (on the left) in Figure 4 is very strange. Specifically, I cannot understand how the deeper layer's soil moisture can fluctuate that much and at the same time the upper layers to have almost stable soil moisture values. It seems like the water is coming and leaving from bellow. Especially the rapid drops of soil moisture in the deeper layer while the top layers have steadily high values is very strange. Can you please check? Is there a huge difference in soil properties of the soil layers? This should be investigated further.
|
We got the point of the Reviewer, and we agree. The graph reporting VWC seasonal variations reports the average of the three monitoring stations that in the case of CONS showed a similar pattern. Moreover, the same trend was seen also in the upper layers of CONV. Also Camarotto et al., (2018) observed the same dynamics in the same monitoring stations. According to Reviewer’s suggestion, we further expanded these findings by including the effect of CONS on pore network and, in turn, on water infiltration (L434-441). |
3. I feel like the obtained results and their analysis were not conclusive but provided mostly indications that could fit with various interpretations. Parts of the results are interesting and some of the findings seem to be in line with previous studies findings, however, the added value of the current study isn't clear. The obtained results from the various monitoring methods are analysed separately and there isn't any synthesis. Accordingly the main conclusions of the study are mostly based on indications and assumptions and not on the analysis of the data. The reality is that there is a good amount of data that with better / deeper analysis could obtain more solid conclusions.
|
We thank the Reviewer for the comment. New considerations have been introduced in the revised text by including the effect of soil covering, evapotranspiration, tillage, bulk density and texture on soil VWC dynamic (L420-447). We added a more quantitative analysis by including new correlations between studied parameters (L329-334). Finally, we revised the entire conclusion section, as suggested (L527-536). |
Reviewer 4 Report
Dear authors,
I have found this manuscript remarkable as it covers an engaging topic as it is the combination of ERT and EMI techniques, supplemented by other additional techniques or analyses, to assist in the assessment of soil water distribution depending on two different agriculture practices. I think this kind of works support the importance of geoelectrical techniques as relevant tools in precision agriculture. The manuscript provides significant findings appropriately supported by the figures and it is well structured, as well. So, potential specialists and/or practitioners involved in agrogeophysics will be pleased to read this manuscript. Personally, I found interesting the Python-related software used for inverting ERT and EMI data, which highlights the actual capabilities of present open-source software to precisely undertake dataset processing avoiding the need to use other costly software. This work demonstrates the usefulness of the ERT and EMI methods to detect those differences between conservation and conventional soil treatments. As the authors suggest, it is a first step forward paving the way for more refined hydrology models to identiy which soil parameters are key to control spatial and temporal changes in soil water content. Moreover, I found the reference list up-to-date and very relevant, which will assist without any doubt other scientists in going deeper in the knowledge of this topic. So, in my opinion, this paper not only properly suits the scope of the journal but also is going to become a must-be-read paper on the topic of precision agriculture and/or agrogeophysics.
Once said the previous comments I would suggest the following:
Page 2, Line #47, please remove one of the spaces and write “physico-chemical” instead of “physio-chemical”.
Page 2, Line #97, please correct “ah inherent”.
Page 5, Line #215, please write “The electro-magnetometry technique” instead of “The electro-magnetometry techniques”.
I suggest the figure order should be kept, so please Fig. 2 must be referenced before Fig. 3 and Fig. 4.
So, I would recommend this manuscript for publication after my minor suggestion has been accomplished.

Author Response
Brief summary
The use of geophysical tools (ERT and EMI) to analyse the impacts of various management approaches (conventional and conservative) on the soil is evaluated in this research, particularly with regard to soil water content and water redistribution along the soil depth.
Broad comments
Comment |
Reply |
Dear authors, I have found this manuscript remarkable as it covers an engaging topic as it is the combination of ERT and EMI techniques, supplemented by other additional techniques or analyses, to assist in the assessment of soil water distribution depending on two different agriculture practices. I think this kind of works support the importance of geoelectrical techniques as relevant tools in precision agriculture. The manuscript provides significant findings appropriately supported by the figures and it is well structured, as well. So, potential specialists and/or practitioners involved in agrogeophysics will be pleased to read this manuscript. Personally, I found interesting the Python-related software used for inverting ERT and EMI data, which highlights the actual capabilities of present open-source software to precisely undertake dataset processing avoiding the need to use other costly software. This work demonstrates the usefulness of the ERT and EMI methods to detect those differences between conservation and conventional soil treatments. As the authors suggest, it is a first step forward paving the way for more refined hydrology models to identiy which soil parameters are key to control spatial and temporal changes in soil water content. Moreover, I found the reference list up-to-date and very relevant, which will assist without any doubt other scientists in going deeper in the knowledge of this topic. So, in my opinion, this paper not only properly suits the scope of the journal but also is going to become a must-be-read paper on the topic of precision agriculture and/or agrogeophysics.
Once said the previous comments I would suggest the following: |
We thank the anonymous Reviewer for the comments that helped us to improve the manuscript quality. |
Specific comments
Page 2, Line #47, please remove one of the spaces and write “physico-chemical” instead of “physio-chemical”.
|
Revised, thank you (L45). |
Page 2, Line #97, please correct “ah inherent”.
|
The entire sentence has been reformulated |
Page 5, Line #215, please write “The electro-magnetometry technique” instead of “The electro-magnetometry techniques”.
|
Revised, thank you (L213). |
I suggest the figure order should be kept, so please Fig. 2 must be referenced before Fig. 3 and Fig. 4.
|
The reference to fig. 3 and fig. 4 in the MM section has been removed |
Round 2
Reviewer 3 Report
The manuscript is somehow improved but in reality the changes are very limited. As regards my previous first comment I agree with the authors. In my previous review I didn't have information about the special issue even if I tried to check as it was strange that the manuscript was submitted to this topic.
My second and third comments are still valid. As regards the strange shape of soil moisture temporal variation the macropores cannot explain these graphs. As regards the third comment the changes in the methodology and the results are very limited and do not really answer to my third comment.
Author Response
Reviewer 2
The manuscript is somehow improved but in reality the changes are very limited. As regards my previous first comment I agree with the authors. In my previous review I didn't have information about the special issue even if I tried to check as it was strange that the manuscript was submitted to this topic.
My second and third comments are still valid.
Specific comments
1. As regards the strange shape of soil moisture temporal variation the macropores cannot explain these graphs.
|
In relation to the soil moisture graph, we came to the conclusions that macropores can explain the quick fluctuations in CONS considering that: i) We exclude it may be an instrumental error since the same dynamic has been observed at all the three stations, as well as documented in the two previous years (Camarotto et al., (2018)). Furthermore, the monitoring probes were calibrated on those soils with an accuracy of ±3%. This is the most suitable interpretation we achieved, also in agreement with literature. However, the authors are willing to accept further interpretation by the reviewer. The text has been revised accordingly (L439-449). |
2. As regards the third comment the changes in the methodology and the results are very limited and do not really answer to my third comment.
|
As previously reported, the discussion section had been improved and reformulated in order to link results from direct and geophysical survey methods, also by adding new correlations. Furthermore, the conclusion section had been totally rewritten, better synthetizing the results reported in the manuscript. To support and validate the geophysical monitoring we performed a detailed statistical assessment of related soil properties as described in the MM section. However, a quantitative geophysical analysis (which would allow a comparison between the methods) would require repeated measurements over multiples field sites or a modelling exercise that cannot be addressed in this manuscript. |